# Comparing level of food insecurity between households with and without home gardening practices in Zege, Amhara region, North West Ethiopia: Community based study

**Achenef Motbainor** [ID]*, **Zerfalem Arega, Mulat Tirfie**

School of Public Health, College of Medicine and Health Sciences, Bahir Dar University, Bahir Dar, Ethiopia

* motbainor2@gmail.com

**Data Availability Statement:** Full data is summited as additional information.

**Funding:** The author(s) received no specific funding for this work.

## Abstract

### Background

Globally, close to 1 billion people suffer from hunger and food insecurity. Evidence showed that prevalence of household food insecurity in Ethiopia is ranged from 25.5%-75.8%. Home gardening is one way to alleviate food insecurity. Hence, the study aimed to determine level of food insecurity and its associated factors between home gardening and non-home gardening household in Zegie, North west Ethiopia.

### Methods

Community-based study was conducted from February 10th-March 10th/2020. A total of 648 samples were included. First, 2142 total households who have 6–59 months of age children in the area identified and registered. Then, households categorized in to home garden practicing (1433) and non-home garden practicing (709). The calculated sample size, 324 for each group were selected using simple random sampling technique.

### Results

The overall prevalence of food insecurity was 38.1% (95% CI: 34.29–42.11%). Food insecurity was significantly higher in non-home gardening groups than their counter parts 45.5% (95% CI: 39.80–51.20%). Having primary education and above (AO = 1.89, 95% CI: 1.25–2.86%), wealth index; 2nd quantile (AOR = 0.44, 95% CI: 0.25–0.85%), 3rd quantile (AOR = 0.32, 95% CI: 0.17–0.62%) and 4th quantile (AOR = 0.27, 95% CI: 0.15–0.54%), dietary diversity (AOR = 1.83, 95% CI: 1.15–2.92%) and home garden practices (AOR = 1.57, 95% CI: 1.06–2.32%) were variables significantly associated with food insecurity.

### Conclusion

Food insecurity in non-home garden practicing households is higher than practicing households. The local agriculture sector needs to emphasis and empowered households on home gardening practices to realize food security.

**Competing interests:** The authors have declared that no competing interests exist.

**Abbreviations:** AOR, Adjusted Odds Ratio; CI, Confidence Interval; EHFP, Enhanced Homestead Food Production; FANTA, Food and Nutrition Technical Assistance; FIAS, Food Insecurity Access Scale; HGP, Homestead Garden Production; HHs, Households; NGOs, Non-Governmental Organizations; PCA, Principal Component Analysis; SD, Standard Deviation; WHO, = World Health Organization.

## Introduction

Food security, which can be descried as the ability of all people in the society to get enough and nutritious food at all times for an active, healthy and productive life, has increasingly been recognized as a serious public health issue over the past 2–3 decades [1–4]. The other way of considering the issue is food insecurity which is highly prevalent in developing countries. Globally, different levels of food insecurity have been registered, for example in Iran it reached up to 49.2% [5, 6]. Presently, close to 1 billion people at global level suffer from hunger and food insecurity [3]. The mild, moderate and severe food insecurity in rural Cambodia was 33%, 37% and 12%, respectively [7]. There are some extreme results from developing countries that stated the magnitude of food insecurity. For example, in Kenya severe food insecurity reached up to 62.5% [8]. Statistics in Kenya also indicated that over 10 million people suffer from chronic food insecurity and poor nutrition; 2–4 million people require emergency food assistance at any given time [3].

Ethiopia is also one of the countries where high level of food insecurity is existing. Evidences from both small-scale and large-scale studies of the different parts of the country showed that household food insecurity prevalence ranged from 25.5% to 75.8% [9–12]. In Gambella town the prevalence of household food insecurity was 59.5% with 20.1%, 23.6%, and 15.8% mild, moderate and severe, respectively [10]. Similarly, study results from east and west Gojjam zones of the Amhara region showed that household food insecurity was 59.2% and 51.3%, respectively [11]. Another evidence from East Badawacho District, South Ethiopia, showed that the prevalence of household food insecurity was 75.8% [12]. The presence of high level of prevalence discrepancies and different intervention actions make further studies necessitate to determine the status of the magnitude and see the changes that happening.

Because it is a complex and multidimensional concept, measuring food insecurity has been an ongoing challenge to researchers and practitioners alike [13]. Despite apparent empirical strength, the operationalization of the food security concept still presents many challenges. Having similar definitions for food security, measurements and assessments methods is difficult and can differ considerably, even within the boundaries of the qualitative and quantitative traditions [14]. The available food security measurement methods considered per capita calorie, number of foods or food groups, proportion of income people spend on food, behavior of people related to food consumption and behavioral and psychological measures in relation with food consumption are the main concepts for measurement [15, 16]. Household Food Insecurity Access Scale (HFIAS), food consumption score, household expenditure survey, calorie intake/food energy availability per capita, and coping strategy are some to mention from the different methods used by different experts to assess food security/insecurity level.

Different factors contribute for the occurrence of food insecurity at household level. Generally, shortage of farmland, poverty, recurrent drought and climate change, shortage of rainfall, and land degradation are well known determinants of food insecurity [17]. However, in different specific areas and studies there are varied determinants of food insecurity. For example, a study done in Iran stated that the presence of chronic disease in household, the presence of smoker in household, the residential area, household income were factors identified [5]. Studies from Ethiopia showed that there are different factors that affect household food security. These factors include; having large family size, those who cannot read and write, and households lead by old household heads. In addition, having small size farm lands, lack of livestock, those who do not have confidence to overcome food insecurity of the households, not using farm inputs by the households are other factors affecting food insecurity [18].

An emerging body of literature links food insecurity to a range of negative health outcomes and causes of a decline in productivity. Household food insecurity is associated with poor

nutritional health [19]. Poor nutritional status has negative consequences on growth and development among children, especially during the early years of life [20]. According to the study results in rural, urban, and peri-urban areas of eight countries of the world, when there is an increase in food access insecurity score, child height-for-age z-score or standard deviation is significantly decrease or when there is an increase in food access insecurity score, the child become more stunted [21].

A study from Brazil investigated the association between food insecurity and children's health and nutrition and its role in the child health income gradient and found that children living in households with food insecurity have worse nutrition and health indicators [22]. Evidence from study result in severely food-insecure households of Nepal showed that 51% of children were stunted compared to children from food secured households [2]. Study from Kenya showed that children from food insecure households are more at risk of becoming stunted to 12 percent than their counter parts. With this same study, the risk of stunting increased by 19% from moderately food insecure households and by 22% from severely food insecure households when the effects of food security and wealth status together assessed [23]. However, there is a controversy in that study result from Kenya stated absence of significant association between food insecurity and stunting [8]. It has also a direct relationship with nutritional status of people associated with self-reported hypertension, hyperlipidemia [24].

Globally, different interventions have been conducted so far to alleviate food insecurity. Home gardening which is explained as a small-scale agricultural production system supplying plant and animal consumption, special care and marked by low capital input and simple technology is the one from the different strategies [25]. Study finding from Kenya showed that home gardening played a significant role in food security of rural households with respect to size of land and food stock and number of livestock and food stock [26]. Other evidence also recommended that intervention strategies should give emphasis to women's education, diversified income generating opportunities, and for different agroecological zone, mixed agriculture strategy [11] which might have a direct relation with home gardening practice of women. It remains the most important method of food production for a majority of people in the developing world [26]. Households, therefore, should be empowered and encouraged to improve their practice of home gardening to realize food security. Based on the evidences that descried home gardening practices as means of reducing food insecurity prevalence at household level, comparing food security status among households who practiced home gardening and who did not is very crucial. Therefore, the current study aimed to conduct community based comparative cross-sectional study to compare proportion of food insecurity between households who practicing and non-practicing of home gardening and the associated factors of food insecurity among these households in Zegie rural satellite town of Bahir Dar city administration.

## Methods and materials

### Study setting and population

Community based comparative cross-sectional study was conducted in Zegie rural satellite town of Bahir Dar city administration, Amhara Regional State, Ethiopia. The study period was from February 10th to March 10th/2020. Zegie rural satellite town is found at a distance of 600 km away from Addis Ababa, capital city of Ethiopia and 32 km from Bahir Dar in the northwest direction of the country. Zegie peninsula is one of the religious area found in the region and households did not have enough farm lands for production of diversified crops [27].

Based on 2019 population projection given from Bahir Dar city administration, the total population size of Zegie was 10,083 (4,041 males and 6,042 females). For administrative

purpose Zegie is divided in to three kebeles (the smallest administrative units of the government). There were 2,142 total households in the town and from theses 709 households practised home gardening while the remaining 1,433 were not. All mothers who have 6–59 months of age children in the household were considered as the source population of the study. This happened due to the fact that the study has other objective that determined the nutritional status of children between the two populations. Selected mothers from the two population groups (households who practised home gardening and who did not) were considered as study population. In the home gardening practised population groups those households started their home gardening practice for at least 6 months were included in the study.

## Sample size determination

The sample size of the study was determined using double population proportion formula by considering the following assumptions; 95% confidence level, 80% power of the study, $P_1$ and $P_2$ the prevalence of stunting in home gardening and non-home gardening populations, respectively.

$$n = \left[ \frac{(Z_1 + Z_2)^2 * 2P(1 - P)}{(P_1 - P_2)^2} \right]$$

$$n = \left[ \frac{(1.96 + 0.84)^2 * 2 * 0.47(1 - 0.47)}{(0.41 - 0.525)^2} \right] = 293.38 \approx 294$$

Where, n = Sample size for each group
$Z_1$ = 1.96 for 95% confidence level, $Z_2$ = 0.84 for 80% power of study
$P = \frac{P_1 + P_2}{2} = \frac{0.41 + 0.525}{2} = 0.47$

$P_1$ = prevalence of stunting in under five children with home gardening practiced households (41%) from previous study [28] and $P_2$ = prevalence of stunting in under five children from households without home gardening practiced (52.5%) [29]. Stunting was considered to estimate the sample size since it is perceived that the nutritional status of the community can be well explained by it than other indices. Other reason for the consideration of stunting for the sample size estimation there was other objective addressed by this study which was to determine the nutritional status of under five children in the study area. Having the above given conditions and 10% non-response rate, the calculated sample size was 648 paired child-mother/care givers. The estimated sample size was checked for its sufficiency by comparing with sample sizes that estimated by considering other factors.

## Sampling technique

First 2142 total households who have 6–59 months of age children in the town were identified and registered at the health post level. This was done to address the previously stated objective, that is to determine under five children nutritional status. Then these households were grouped in to home gardening (1433) and non-home gardening (709). Then, the calculated sample size (648), 324 for each group were selected using simple random sampling (computer generating method) technique.

## Study variables

**Dependent variable.** Food security/insecurity
**Independent variables.** Socio-demographic variables (marital status, maternal education, paternal education, family size, household head, occupational status), wealth index.

## Operational definitions

**Home gardening.** Is defined as households who cultivate at least one kind of fruit and vegetable in their yard or compound.

**Higher dietary diversity score.** Defined as proportion of households who receive 4 or more food groups from the 7 food groups consumed over 24 hours [30].

**Low dietary diversity score.** Defined as proportion of households who receive 3 or less food groups from the 7 food groups consumed over 24 hours [31]

**Food secure households.** Defined as households experiences none of the food insecurity (access) conditions, or just experience worry, but rarely otherwise food insecure households [13].

**Land size.** Is defend as the physical size of the farm, primarily in terms of hectares of operated land [32].

**Decision-making power of women.** A woman said to be decision maker if she participated lonely or and jointly 5 and above from 10 decision making related questions [33].

## Data collection tools and procedures

Different types of tools and measurements were implemented to collect the required data. Structured interviewer administered questionnaire was developed by reviewing different literature. The questionnaire was developed in English and translated to the local language (Amharic) and back to English to check its consistency. The questionnaire has sections like socio-demographic, and/or socio-economic characteristics, nutrition related, wash related, health related factors and anthropometric measurements. After households who have 6–59 months of age children selected from health posts, then data collectors went to the house for interview. Four clinical nurses and two health officers were assigned for data collection and supervisory respectively.

**Dietary diversity of children.** A 24-hour recall method (from sun rise to sun rise) was used to assess dietary diversity practices. It was based on the mother's recall of foods given to her child in the previous 24 hours prior to the interview date. Then, minimum dietary diversity was estimated using information collected from the 24 hours dietary recall. Minimum dietary diversity was fulfilled if a child had received four or more food groups from the seven WHO food groups in the last 24 hours preceding the survey. Seven food groups included were grains, roots, and tubers; legumes and nuts; dairy products (milk, yogurt, and cheese); flesh foods (meat, fish, poultry, and liver/organ meats); eggs; vitamin rich fruits and vegetables; and other fruits and vegetables [30].

**Food security status of the households.** Household food-security (access) information was collected by using the questionnaire adopted from the Household Food Insecurity Access Scale, which was developed by the Food and Nutrition Technical Assistance project. This instrument consists of nine questions that measure uncertainty on obtaining food, limited access to high-quality foods, and reduction in food quantity in the past 4 weeks. The precoded options were never (0 points), rarely (once or twice in the past 4 weeks; 1 point), sometimes (three to ten times in the past 4 weeks; 2 points), and often (more than ten times in the past 4 weeks; 3 points). Scores for answers to these questions were summed (0–27), and thus a household experiences none of the food insecurity conditions, or just experiences worry, but rarely categorized as food secure otherwise food insecure household [13].

**Wealth index of the households** was determined using the Principal Component Analysis (PCA). Communality value > 0.5, KMO (sampling adequacy) with P-value > 0.05 and complex structure factor (Eugene value) greater than 1 was considered. Quintiles of the wealth score was created to categorize households as poorest (1st quantile), poor (2nd quantile), medium (3rd quantile), rich (4th quantile) and richest (5th quantile).

## Data quality control

To maintain the quality of data, first, standardized data collection tools were adopted from published sources and contextualized to the local study area. Pretest was done on 5% of the total sample size (26) other than study sites with similar characteristics. Weighing scale was calibrated before each measurement using known weight and all anthropometric measurements were taken twice, and the average of the two measurements were calculated and recorded. Two days training was given for data collectors and supervisors prior to the actual data collection time on the selection procedure of study participants, purpose of the study, on the steps how they can give the necessary information for the participants when they start data collection. The supervisor and principal investigator were supervised and checked the completeness and quality of data daily. During data collection, questionnaires were reviewed and checked for completeness by the supervisor and principal investigator and the necessary feedback was offered to the data collectors in the next morning. Then the data obtained from the study population were entered, and cleaned for missing value by the investigator.

## Data processing and analysis

The collected data was coded, entered and cleaned using Epi data version 3.02 and exported to SPSS version 23 for analysis. Descriptive statistics like frequency, percentage and mean were carried out for different variables. The association between two populations was cheeked using chi square test. Bi variable logistic regression analysis was used to know the crude association between each independent variables and outcome variable (stunting and wasting) and crude odds ratio was taken. Then variables which were associated with the dependent variable in bi-variable analysis with p-value < 0.2 were included in the models of multivariable logistic regression analysis with backward likelihood ratio approach. The Hosmer–Lemeshow test was performed for model fitness in the final model, and p-value > 0.05 was considered as a good fit. Anthropometric data were converted in to indices and indicators using WHO Anthro software.

Having p-value less than 0.05 in multivariable logistic regression analysis was used to conclude the presence of statistically significant association between different predictor variables with outcome variable (stunting and wasting). The strength of statistically association was measured by adjusted odds ratio at 95% confidence level.

## Results

### Socio demographic characteristics of study participants

A total of 648 mothers, (with 100% response rate) participated in the study. Almost sixty eight percent (67.9%) of mothers from households with home garden and eighty percent (79.9%) of mothers from households without home garden were with no formal education. The mean (+SD) age of the mothers was 28.7 (±5.5) years in households with home garden, and 29.7 (±5.8) years in households without home garden. Nearly, all of the mothers in both (with home gardening 99.4% and non-home gardening 98.4%) households were married. The majority of the households were male headed: 242 (78.6%) in households with home garden and 219 (71.1%) in households non-home gardening. The mean (±SD) family size in this study was 4.22±1.069 from HHs with home garden and 4.29±0.941 from HHs non-home gardening. About 262 (85.1%) and 246 (79.9%) respondents lived in rural settlements from HHs with and non-home gardening, respectively (Table 1). Considering the wealth index of the two population groups, almost sixty percent (61.0%) of the people found in the fifth quantile of the non-home gardening whereas for home gardening only 39.0% found in the fifth quantile and the larger percent of this group (59.0%) found in the first quantile (Table 1).

**Table 1. Socio-demographic and economic characteristics of home gardening and non-home gardening households in Zegie, Northwest Ethiopia, 2019.**

| Variables | Category | Home gardening households | Non-home gardening households | Chi-square (p-value) |
|---|---|---|---|---|
| | | Frequency (%) | Frequency (%) | |
| Maternal age | 20–29 years | 178 (54.1) | 151 (45.9) | 4.999(0.082) |
| | 30–39 years | 122 (45.7) | 145 (54.3) | |
| | > 30 years | 8 (40.0) | 12 (60.0) | |
| Maternal education | No formal education | 209 (67.9) | 190 (61.7) | 10.639(0.001) |
| | Primary and above | 99 (32.1) | 118 (38.3) | |
| Father education | No formal education | 209 (33.4) | 109 (36.8) | 0.746 (0.388) |
| | Primary and above | 99 (66.6) | 187 (63.2) | |
| Maternal occupation | House wife | 272 (88.3) | 246 (79.9) | 8.795(0.032) |
| | Merchant | 18 (5.8) | 27 (8.8) | |
| | Employed | 10 (3.2) | 22 (7.1) | |
| | Student | 8 (2.6) | 13 (4.2) | |
| Father occupation | Farmer | 230 (72.2.) | 187 (63.2) | 15.203(0.001) |
| | Merchant | 23 (7.5) | 28 (9.5) | |
| | Employed | 45 (14.6) | 81 (27.4) | |
| Household family size | < 5 | 200 (64.8) | 191 (62) | 0.522(0.47) |
| | ≥ 5 | 108 (35.2) | 117 (38) | |
| Women decision making | Yes | 244 (79.2) | 186 (60.4) | 25.909(0.001) |
| | No | 64 (20.8) | 122 (39.6) | |
| Egg farming of the household | Yes | 87 (28.2) | 81 (26.3) | 0.295(0.587) |
| | No | 221 (71.8) | 227 (73.7) | |
| Dairy farming of the household | Yes | 135 (43.8) | 124 (40.3) | 0.806(0.369) |
| | No | 173 (56.2) | 184 (59.7) | |
| Bee farming of the household | Yes | 27 (8.8) | 11 (3.6) | 7.18(0.007) |
| | No | 281 (91.2) | 297 (96.4) | |
| Feeding frequency | ≥ 3 times per day | 148 (48.1) | 124 (40.3) | 3.792(0.051) |
| | < 3 times per day | 160 (51.9) | 184 (59.7) | |
| Dietary diversity score | High | 208 (67.5) | 106 (34.4) | 67.584(0.001) |
| | Low | 100 (32.5) | 202 (65.6) | |
| Wealth index | First quantile | 82 (59.0) | 57 (41.0) | 17.697(0.001) |
| | Second quantile | 60 (53.6) | 52 (46.4) | |
| | Third quantile | 67 (57.3) | 50 (42.7) | |
| | Forth quantile | 51(40.8) | 74 (59.2) | |
| | Fifth quantile | 48 (39.0) | 75(61.0) | |

## Level of food insecurity

The overall prevalence of food insecurity in the study area found to be 38.1% (95%CI: 34.29, 42.11%). Comparing the two population groups, food insecurity found to be significantly higher in nonhome gardening groups than their counter parts with prevalence of 45.5% (95% CI: 39.80, 51.20%) (Table 2).

**Table 2. Level of food insecurity among home gardening and non-home gardening households in Zegie, Northwest Ethiopia, 2019.**

| Home gardening practices | Food security status | | | |
|---|---|---|---|---|
| | Secure (Frequency) | Prevalence (95% CI) | Insecure (Frequency) | Prevalence (95% CI) |
| Yes | 213 | 69.16% (63.67, 74.27) | 95 | 30.84% (25.73, 36.33) |
| No | 168 | 54.5% (48.80, 60.20) | 140 | 45.5% (39.80, 51.20) |
| Total | 381 | 61.9% (57.88, 65.70) | 235 | 38.1% (34.29, 42.11) |

## Factors associated with food insecurity

All variables were checked for their association with food security during the bivariate analysis and those variables associated with p-value $\leq 0.25$ significance level considered as candidate variables for multivariable logistic regression and taken to the next step.

Maternal education status, wealth index, dietary diversity and home gardening practices of the household were factors found to be significantly associated with food security. The odds of mothers who had primary and above education level was 1.89 times higher to be food secure (AOR = 1.89, 95% CI: 1.25, 2.86) than mothers who did not attended education. The 2nd, 3rd and 4th quantiles of wealth index of the households was also found to be food insecure compared to the 5th quantile. For example, the odds of the households in the 2nd quantile of the wealth index was 0.44 times lower to be food secure (AOR = 0.44, 95% CI: 0.25, 0.85) compared to the households found in the 5th quantile. In other words, 56% of the households found in the 2nd quantile of the wealth index was food insecure compared with households found in the 5th quantile of the wealth index. In the same way, the odds of the households in the 3rd quantile was 0.32 times lower to be food secure (AOR = 0.32, 95% CI: 0.17, 0.62) compared to the households found in the 5th quantile. Households with high food diversity and practicing home gardening were also food secure compared their counter parts. The odds of households who practicing home gardening was 1.57 times higher to be food secure (AOR = 1.57, 95% CI: 1.06, 2.32) compared with households who did not practicing home gardening (Table 3).

**Table 3. Logistic regression analysis showed factors significantly associated with food security among home gardening and non-home gardening households in Zegie, Northwest Ethiopia, 2019.**

| Variables | Category | Food secured N (%) | Food insecure N (%) | COR (95% CI) | AOR (9% CI) |
|---|---|---|---|---|---|
| Maternal education | Primary and above | 153 (70.5) | 64 (29.5) | 1.79 (1.26, 2.55)*** | 1.89(1.25, 2.86)*** |
| | No education | 228 (57.1) | 171(42.9) | 1 | 1 |
| Father education | Primary and above | 244 (63.2) | 142 (36.8) | 0.87(0.61, 1.22) | 0.84(0.57, 1.22) |
| | No education | 125 (59.8) | 84 (40.2) | 1 | 1 |
| Father occupation | Farmer | 265 (63.5) | 152 (36.5) | 0.74(0.49, 1.11) | 0.77(0.47, 1.28) |
| | Merchant | 32 (62.7) | 19 (37.3) | 0.77(0.39, 1.49) | 0.61(0.29, 1.30) |
| | Employed | 71(56.3) | 55 (43.7) | 1 | 1 |
| Wealth index | 1st quantile | 74 (53.2) | 65(46.8) | 0.84(0.53, 1.41) | 0.73(0.39, 1.36) |
| | 2nd quantile | 75 (67.0) | 37 (33.0) | 0.48(0.28, 0.82)** | 0.44(0.25, 0.85)* |
| | 3rd quantile | 84 (71.8) | 33(2.2) | 0.39(0.23, 0.66)*** | 0.32(0.17, 0.62)*** |
| | 4th quantile | 87(69.6) | 38 (30.4) | 0.43(0.26, 0.72)*** | 0.27(0.15, 0.54)*** |
| | 5th quantile | 61(49.6.) | 62 (50.4) | 1 | 1 |
| Food diversity | High | 227(72.3) | 87(27.7) | 2.51(1.79, 3.50)*** | 1.83(1.15, 2.92)** |
| | Low | 154(51.0) | 148(49.0) | 1 | 1 |
| Egg farming | Yes | 122(72.6) | 46(27.4) | 1.94(1.31, 2.85)*** | 1.36(0.85, 2.17) |
| | No | 259(57.8) | 189(42.2) | 1 | 1 |
| Dairy farming | Yes | 180(69.5) | 79(70.5) | 1.77(1.26, 2.48)*** | 1.27(0.84, 1.93) |
| | No | 201(56.3) | 156(43.7) | 1 | 1 |
| Home gardening practice | Yes | 213(69.2) | 95(30.8) | 1.87(1.34,2.59) *** | 1.57(1.06, 2.32)* |
| | No | 168(54.5) | 140(45.5) | 1 | 1 |

* = P-value < 0.05

** = P-value < 0.01

*** = P-value < 0.001

Whereas egg farming and dairy farming activities of the households were found to be statistically significant variables associated with food security in the bivariate analysis but its significant association vanished in the final model or multivariable logistic regression (Table 3).

## Discussion

Following the Helen Keller International pioneered on enhanced homestead food production (EHFPs) programs, many other non-governmental organizations (NGOs) have adopted homestead garden production (HGPs) or EHFPs in different settings [34]. There is an agreement so far where agricultural growth by itself does not necessarily bring about family nutrition improvements. Home gardening practice is believed to have great role in family nutrition improvements and has positive results. Hence, responsible agents like governments, donors, UN agencies and NGOs need to invest to promoting it [35]. The agriculture sector of Federal Democratic Republic of Ethiopia tried to adopt this homestead or home gardening agricultural practices as means of addressing food insecurity problems focusing to the urban communities. The current study focused on the rural community to compare their level of food security between home gardening and non-home gardening households.

The study found that food security significantly differs between the two groups where households practicing home garden were more food secure than those who not practicing it. The overall prevalence of food insecurity in this study was lower than other findings done in different parts of the country that include Gambela, East Badawacho District, South Ethiopia, East and West Gojjam zones of Amhara Region [10–12]. This difference might be resulted from the presence of this home garden practices in the study area. This is in line with the finding of this current study in that food insecurity in home garden practicing households found to be lower than their counter parts. The finding was also similar with findings from other parts of the world such as Nepal, Indonesia and China [2, 36, 37]. Generally, there is a promising positive improvement in their food security status of the households who practicing home garden. This positive improvement towards household food security among households who practiced home garden was also reported in other study [38].

Besides the prevalence of food insecurity, the study was also tried to identify factors associated with food security in the area. Hence, maternal education status, wealth index of the households, dietary diversity and home gardening practice were factors significantly associated with food security status of the households. Households with mothers who had primary and above educational status were more food secure than their counter parts. This could be explained in that educated mothers are at the better position to properly implement the intervention packages and advises given for those who are practicing home garden. The issue is ensured by study reports from Ethiopia and south Africa where education positively affect the food security status of households [39, 40]

As it is expected, household wealth index was also showed significance association to household food security. Households within the 1$^{st}$, 2$^{nd}$, and 3$^{rd}$, quantiles of wealth index found to be less likely to be food secure compared the last quantile. Although it is not always true, this might be due to better economic status of households in the last quantile of wealth indexes and could easily accessed the required foods. Other study in Ethiopia supported this finding in that when the wealth index of the household increased being food insecure decreased [39].

The other significantly associated factors with food security were dietary diversity and home gardening practices of households. Home gardening contributes to household food security by providing direct access to food that can be harvested by households, this harvested foods in turn can also prepared and fed to family members, often on a daily basis. Evidence reveled that home gardening practice positively affect households living standard by

improving household food security status and household income [41]. Another study result obtained from Kenya showed that home gardening plays a significant role in food security of rural households [26]. Generally, home gardening improved household food security in different ways that can be described as improving household income or economic status, the type of food items accessed by the household members which can be called dietary diversity and empowering women to decide on harvested foods [42–44]. Hence, households, need be empowered and encouraged to improve their practice of home gardening to realize food security.

## Limitation of the study

This study has a limitation of not including different factors that include access to water, knowledge or skill differences related to home gardening practice, different inputs and time or seasonal variations which may have potential impacts on both home gardening practices and food security status.

## Conclusion

Comparing to other areas of the country, the finding of this study showed lower prevalence of food insecurity. Home gardening showed a promising positive effect to reduce food insecurity among rural households of the study area. Besides home gardening dietary diversity showed statistically significant association with food security. Home gardening practices of the households should be enhanced in the rural community since it has improving dietary diversity, women empowerment in decision making on home garden products and economical effects.

## Supporting information

**S1 File.**
(SAV)

## Acknowledgments

Authors want to acknowledge Bahir Dar University for the support it provided us data collection budget. We want also to acknowledge data collectors, supervisors and participants for their time and information they provided.

## Author Contributions

**Conceptualization:** Achenef Motbainor, Zerfalem Arega.

**Data curation:** Achenef Motbainor, Zerfalem Arega, Mulat Tirfie.

**Formal analysis:** Achenef Motbainor, Zerfalem Arega, Mulat Tirfie.

**Investigation:** Zerfalem Arega, Mulat Tirfie.

**Methodology:** Achenef Motbainor, Zerfalem Arega, Mulat Tirfie.

**Writing – original draft:** Achenef Motbainor, Mulat Tirfie.

**Writing – review & editing:** Achenef Motbainor.

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
