## [Decision Letter · Decision Letter 0]

19 Jul 2022

PONE-D-22-10969Comparing level of food insecurity between households with and without home gardening practices in Zege, Amhara region, North West Ethiopia: Community based studyPLOS ONE

Dear Dr. Motbainor,

Thank you for submitting your manuscript to PLOS ONE. After careful consideration, we feel that it has merit but does not fully meet PLOS ONE’s publication criteria as it currently stands. Therefore, we invite you to submit a revised version of the manuscript that addresses the points raised during the review process.

We look forward to receiving your revised manuscript.

Kind regards,

Stefano Marchetti, Ph.D.

Academic Editor

PLOS ONE

Journal Requirements:

- https://www.eajournals.org/wp-content/uploads/Contribution-of-Home-Gardening-to-Family-Food-Security-in-Delta-North-Agricultural-Zone-Delta-State-Nigeria.pdf 

- http://www.iwmi.cgiar.org/Publications/Working_Papers/working/wor162.pdf

- https://www.ccsenet.org/journal/index.php/jas/article/viewFile/27277/17938

- https://www.science.gov/topicpages/h/household+food+access

- https://cyberleninka.org/article/n/1338744

- https://agricultureandfoodsecurity.biomedcentral.com/articles/10.1186/s40066-019-0254-0

In your revision ensure you cite all your sources (including your own works), and quote or rephrase any duplicated text outside the methods section. Further consideration is dependent on these concerns being addressed

Additional Editor Comments (if provided):

The paper has been revised by two reviewers, who suggested major revision. Please, replay to any issues raised by the reviewers before submitting the revised paper.

Reviewers' comments:

Reviewer's Responses to Questions

**Comments to the Author**

1. Is the manuscript technically sound, and do the data support the conclusions?

Reviewer #1: Yes

Reviewer #2: Yes

2. Has the statistical analysis been performed appropriately and rigorously? 

Reviewer #1: Yes

Reviewer #2: Yes

3. Have the authors made all data underlying the findings in their manuscript fully available?

Reviewer #1: No

Reviewer #2: No

4. Is the manuscript presented in an intelligible fashion and written in standard English?

Reviewer #1: No

Reviewer #2: No

5. Review Comments to the Author

Reviewer #1: Manuscript ID: PONE-D-22-10969

Comparing level of food insecurity between households with and without home

gardening practices in Zege, Amhara region, North West Ethiopia: Community based

study

This paper deal with the food insecurity of household located in Zegie, North west Ethiopia. The aim of the paper is to determine level of food insecurity and its associated factors between home gardening and non-home gardening household. Achieving food security is a significant and growing challenge in the developing countries and highly critical to alleviating poverty.

Results reveal that food insecurity in non-home garden practicing households is higher than practicing households.

The topic is very interesting and may have relevant policy implications, both at the firm and national level. I consider the paper connected with the overall philosophy of the Journal. Statistical analyses are performed to a technical standard and are described in sufficient detail. Conclusions are presented in an appropriate fashion and are supported by the data.

However, the current version of the paper suffers from a number of weaknesses. In what follows, I shall try to point out my main concerns.

I would like to know how the dependent variable (food security/insecurity) has been identified?

In literature there are several indicators for monitoring household poverty levels, food poverty and access to food. Authors should better explain this point in a section devoted to the literature review.

Moreover, the aim of the paper and the strategy followed in the paper should be better identified in the introduction section.

Finally, I suggest authors to provide a proofread of the paper since there are some typos (eg. In line 44 “developing counties” should be “developing countries”.

In general, I would suggest to accepts the paper with major revisions.

Reviewer #2: Comparing level of food insecurity between households with and without home

gardening practices in Zege, Amhara region, North West Ethiopia: Community based

study

The article studies food insecurity of household in Zegie, North west Ethiopia. The aim of the paper is to determine level of food insecurity and its associated factors between home gardening and non-home gardening household. Authors find that food insecurity in non-home garden practicing households is higher than practicing households.

The topic discussed in the paper is of interest, because food insecurity is an important aspect of the poverty phenomena, both in developed and developing countries.

The paper as it is need to be revised to be considered for publication in Plos One.

- The authors have to precisely define the food security/insecurity variable.

- The introduction section has to be revised, there is need to describe clearly the aim of the paper and the methods and data used to monitor poverty levels, food poverty and access to food.

- English have to be revised.

6. PLOS authors have the option to publish the peer review history of their article (what does this mean?). If published, this will include your full peer review and any attached files.

Reviewer #1: No

Reviewer #2: No

---

## [Author Response · Author response to Decision Letter 0]

31 Aug 2022

We addressed all the given comments from the editor and reviewers and responded using point-by-point responses.

---

## [Decision Letter · Decision Letter 1]

11 Oct 2022

PONE-D-22-10969R1Comparing level of food insecurity between households with and without home gardening practices in Zege, Amhara region, North West Ethiopia: Community based studyPLOS ONE

Dear Dr. Motbainor,

Thank you for submitting your manuscript to PLOS ONE. After careful consideration, we feel that it has merit but does not fully meet PLOS ONE’s publication criteria as it currently stands. Therefore, we invite you to submit a revised version of the manuscript that addresses the points raised during the review process.

 Please submit your revised manuscript by Nov 25 2022 11:59PM. If you will need more time than this to complete your revisions, please reply to this message or contact the journal office at plosone@plos.org. Please include the following items when submitting your revised manuscript:A rebuttal letter that responds to each point raised by the academic editor and reviewer(s). You should upload this letter as a separate file labeled 'Response to Reviewers'.A marked-up copy of your manuscript that highlights changes made to the original version. You should upload this as a separate file labeled 'Revised Manuscript with Track Changes'.An unmarked version of your revised paper without tracked changes. You should upload this as a separate file labeled 'Manuscript'.If applicable, we recommend that you deposit your laboratory protocols in protocols.io to enhance the reproducibility of your results. Protocols.io assigns your protocol its own identifier (DOI) so that it can be cited independently in the future. For instructions see: https://journals.plos.org/plosone/s/submission-guidelines#loc-laboratory-protocols. Additionally, PLOS ONE offers an option for publishing peer-reviewed Lab Protocol articles, which describe protocols hosted on protocols.io. Read more information on sharing protocols at https://plos.org/protocols?utm_medium=editorial-email&utm_source=authorletters&utm_campaign=protocols.

We look forward to receiving your revised manuscript.

Kind regards,

Stefano Marchetti, Ph.D.

Academic Editor

PLOS ONE

Journal Requirements:

Additional Editor Comments (if provided):

The paper has greatly improved after major revision. Please, follow further recommendations from one of the reviewer.

Reviewers' comments:

Reviewer's Responses to Questions

**Comments to the Author**

1. If the authors have adequately addressed your comments raised in a previous round of review and you feel that this manuscript is now acceptable for publication, you may indicate that here to bypass the “Comments to the Author” section, enter your conflict of interest statement in the “Confidential to Editor” section, and submit your "Accept" recommendation.

Reviewer #1: All comments have been addressed

Reviewer #2: All comments have been addressed

2. Is the manuscript technically sound, and do the data support the conclusions?

Reviewer #1: Yes

Reviewer #2: Yes

3. Has the statistical analysis been performed appropriately and rigorously? 

Reviewer #1: Yes

Reviewer #2: Yes

4. Have the authors made all data underlying the findings in their manuscript fully available?

Reviewer #1: Yes

Reviewer #2: Yes

5. Is the manuscript presented in an intelligible fashion and written in standard English?

Reviewer #1: No

Reviewer #2: Yes

6. Review Comments to the Author

Reviewer #1: the manuscript has improved, but I suggest to the authors to include in the introduction the AIM and SCOPE of this paper, with brief description of the method used (5-6 lines in total) and a summary of the paper.

Reviewer #2: (No Response)

7. PLOS authors have the option to publish the peer review history of their article (what does this mean?). If published, this will include your full peer review and any attached files.

Reviewer #1: No

Reviewer #2: No

---

## [Decision Letter · Decision Letter 2]

6 Dec 2022

Comparing level of food insecurity between households with and without home gardening practices in Zege, Amhara region, North West Ethiopia: Community based study

PONE-D-22-10969R2

Dear Dr. Motbainor,

We’re pleased to inform you that your manuscript has been judged scientifically suitable for publication and will be formally accepted for publication once it meets all outstanding technical requirements.

Kind regards,

Stefano Marchetti, Ph.D.

Academic Editor

PLOS ONE

Additional Editor Comments (optional):

After the second round of revision by one referee I am now happy to inform you that the paper can be accepted for publication on Plos One.

Reviewers' comments:

Reviewer's Responses to Questions

**Comments to the Author**

1. If the authors have adequately addressed your comments raised in a previous round of review and you feel that this manuscript is now acceptable for publication, you may indicate that here to bypass the “Comments to the Author” section, enter your conflict of interest statement in the “Confidential to Editor” section, and submit your "Accept" recommendation.

Reviewer #1: All comments have been addressed

2. Is the manuscript technically sound, and do the data support the conclusions?

Reviewer #1: Yes

3. Has the statistical analysis been performed appropriately and rigorously? 

Reviewer #1: Yes

4. Have the authors made all data underlying the findings in their manuscript fully available?

Reviewer #1: Yes

5. Is the manuscript presented in an intelligible fashion and written in standard English?

Reviewer #1: Yes

6. Review Comments to the Author

Reviewer #1: The manuscript is improved and all the concern highlighted in the previous revision run have been addressed in the revised paper. I have no further suggestions for the authors.

I suggest to accept the paper in the current version.

7. PLOS authors have the option to publish the peer review history of their article (what does this mean?). If published, this will include your full peer review and any attached files.

Reviewer #1: No

---

## [Editor Report · Acceptance letter]

12 Dec 2022

PONE-D-22-10969R2 

Comparing level of food insecurity between households with and without home gardening practices in Zege, Amhara region, North West Ethiopia: Community based study 

Dear Dr. Motbainor:

I'm pleased to inform you that your manuscript has been deemed suitable for publication in PLOS ONE. Congratulations! Your manuscript is now with our production department. 

Kind regards, 

on behalf of

Dr. Stefano Marchetti 

Academic Editor

PLOS ONE